# Impact of Dexamethasone and Inhaled Nitric Oxide on Severe Acute Kidney Injury in Critically Ill Patients with COVID-19

**DOI:** 10.3390/jcm11206130

**Published:** 2022-10-18

**Authors:** Mickaël Bobot, David Tonon, Noémie Peres, Christophe Guervilly, Flora Lefèvre, Howard Max, Youri Bommel, Maxime Volff, Marc Leone, Alexandre Lopez, Pierre Simeone, Julien Carvelli, Sophie Chopinet, Sami Hraiech, Laurent Papazian, Lionel Velly, Jérémy Bourenne, Jean-Marie Forel

**Affiliations:** 1Centre de Néphrologie et Transplantation Rénale, Hôpital de la Conception, AP-HM, 13005 Marseille, France; 2Aix Marseille University, INSERM 1263, INRAE 1260, C2VN, European Center for Medical Imaging Research (CERIMED), Campus Santé Timone, 13005 Marseille, France; 3Assistance Publique—Hôpitaux de Marseille, Hôpital Nord, Médecine Intensive Réanimation, Centre d’Etudes et de Recherches sur les Services de Santé et Qualité de vie EA 3279, Aix-Marseille University, 13015 Marseille, France; 4Département d’Anesthésie-Réanimation, Aix-Marseille University, CHU Conception, AP-HM, 13005 Marseille, France; 5Département d’Anesthésie-Réanimation, Aix-Marseille University, CHU Timone, AP-HM, 13005 Marseille, France; 6Service d’Anesthésie-Réanimation, Hôpital Nord, AP-HM, 13005 Marseille, France; 7CNRS, Institut des Neurosciences de la Timone, UMR7289, 13005 Marseille, France; 8Service de Réanimation et Surveillance Continue, Hôpital de la Timone, AP-HM, Aix-Marseille University, 13005 Marseille, France; 9Department of Digestive Surgery and Liver Transplantation, Hôpital la Timone, AP-HM, 13005 Marseille, France; 10European Center for Medical Imaging Research CERIMED, Laboratoire d’imagerie Interventionnelle Ex-périmentale (LIIE), Aix-Marseille Université, Campus Santé Timone, 13005 Marseille, France

**Keywords:** acute kidney injury, intensive care, dexamethasone, inhaled nitric oxide, COVID-19, ARDS

## Abstract

Background: Acute kidney injury (AKI) is the second most frequent condition after acute respiratory distress syndrome (ARDS) in critically ill patients with severe COVID-19 and is strongly associated with mortality. The aim of this multicentric study was to assess the impact of the specific treatments of COVID-19 and ARDS on the risk of severe AKI in critically ill COVID-19 patients. Methods: In this cohort study, data from consecutive patients older than 18 years admitted to 6 ICUs for COVID-19-related ARDS requiring invasive mechanical ventilation were included. The incidence and severity of AKI, defined according to the 2012 KDIGO definition, were monitored during the entire ICU stay until day 90. Patients older than 18 years admitted to the ICU for COVID-19-related ARDS requiring invasive mechanical ventilation were included. Results: 164 patients were included in the final analysis; 97 (59.1%) displayed AKI, of which 39 (23.8%) had severe stage 3 AKI, and 21 (12.8%) required renal replacement therapy (RRT). In univariate analysis, severe AKI was associated with angiotensin-converting enzyme inhibitors (ACEI) exposure (*p* = 0.016), arterial hypertension (*p* = 0.029), APACHE-II score (*p* = 0.004) and mortality at D28 (*p* = 0.008), D60 (*p* < 0.001) and D90 (*p* < 0.001). In multivariate analysis, the factors associated with the onset of stage 3 AKI were: exposure to ACEI (OR: 4.238 (1.307–13.736), *p* = 0.016), APACHE II score (without age) (OR: 1.138 (1.044–1.241), *p* = 0.003) and iNO (OR: 5.694 (1.953–16.606), *p* = 0.001). Prone positioning (OR: 0.234 (0.057–0.967), *p* = 0.045) and dexamethasone (OR: 0.194 (0.053–0.713), *p* = 0.014) were associated with a decreased risk of severe AKI. Conclusions: Dexamethasone was associated with the prevention of the risk of severe AKI and RRT, and iNO was associated with severe AKI and RRT in critically ill patients with COVID-19. iNO should be used with caution in COVID-19-related ARDS.

## 1. Background

Acute respiratory distress syndrome (ARDS) is the most common complication in critically ill patients with severe COVID-19 [1]. However, COVID-19 is also associated with other organ impairments, such as acute kidney injury (AKI). AKI is the second most frequent condition after respiratory failure during COVID-19, especially in critically ill patients [2], ranging from 20% to 80% of patients admitted to intensive care units (ICU) and is strongly associated with ICU mortality [3]. Various COVID-19-related kidney impairments were described [4], but the most frequent feature is acute tubular necrosis with interstitial inflammatory involvement, followed by glomerular patterns such as focal segmental glomerulosclerosis [5].

Since the onset of the pandemic, numerous specific treatments, such as antivirals or anti-inflammatory drugs, have been tried for severe COVID-19. Corticosteroids are widely used in critical care for their anti-inflammatory and immunosuppressive properties and have proven efficiency in various conditions, such as anaphylaxis, septic shock, exacerbation of asthma or post-aggressive pulmonary fibrosis [6]. There are many types of corticosteroids used in clinical practice with varying glucocorticoid and mineralocorticoid effects. Since July 2020 and the results of the RECOVERY study, dexamethasone has been shown to reduce mortality in COVID-19 patients [7].

In addition, other specific treatments for ARDS, such as inhaled nitric oxide (iNO), prone positioning and neuromuscular blockers, are also used in COVID-19-related ARDS. However, their impact on kidney function was not assessed in COVID-19 patients. Identifying the impact of these specific treatments on the risk of AKI is then critical to improving the outcomes of patients with severe COVID-19.

The aim of our multicentric study was to assess the impact of these treatments on the risk of severe AKI in critically ill COVID-19 patients.

## 2. Methods

Data from a cohort of patients hospitalized in 6 ICUs in the University Hospital of Marseille, France [8] were retrospectively analysed. The inclusion criteria were patients aged 18 years or more hospitalized in the ICU for COVID-19-related ARDS requiring mechanical ventilation (MV). Patients with stage 5 chronic kidney disease (CKD) (kidney failure), i.e., patients with baseline Glomerular Filtration Rate (GFR) below 15 mL/min/1.73 m^2^, those who did not require invasive MV or deceased during the first 24 h were excluded.

Respiratory variables were recorded during the first day following invasive MV. The ARDS severity was staged according to the Berlin criteria [9]. Initial severity at admission in the ICU was assessed by the APACHE-II score.

The follow-up was performed until day 90 (D90). The kidney function and the incidence and severity of AKI were monitored during the entire ICU stay. AKI was defined and staged according to the 2012 KDIGO definition [10]. We chose to focus on stage 3 AKI, which has the most important negative impact on mortality, and because the less severe AKI can be associated with many confounders.

Exposure to renin-angiotensin system inhibitors, such as angiotensin receptor blockers (ARB) or angiotensin-converting enzyme inhibitors (ACEI), non-steroidal anti-inflammatory drugs (NSAID), and specific treatments of COVID-19 and ARDS were systematically monitored.

Quantitative values were expressed as means ± standard derivation, except for the length of stay expressed in medians [interquartile range], and compared with the Student’s t-test or Mann-Whitney U test as appropriate. Qualitative values are expressed as n (%) and compared with chi^2^ or Fisher test as appropriate, followed by Benjamini-Hochberg False Discovery Rate (FDR) controlling procedure for multiple testing. All tests were two-tailed. A *p*-value below 0.05 was considered significant. The multivariate analysis was performed by logistic regression step-by-step. Variables associated with *p* < 0.20 in univariate analysis were included in the multivariate analysis; we also included positive end-expiratory pressure (PEEP) and the need for veno-venous extra-corporeal membrane oxygenation (VV-ECMO), as they were frequently associated with AKI in previous studies [11,12] and to better take into account the respiratory gravity.

The data included in this study were anonymized, and the study was approved according to General Data Protection Regulation and registered at the Health Data Portal and Data Protection Commission of the AP-HM (reference N°2020-53).

This study was approved by the Ethics Committee of the French Society of Anaesthesia and Intensive Care Medicine (SFAR) (N° 00010254-2020-06) and was exempted from informed consent requirements owing to its retrospective design. The patients were informed and had the option to withdraw their health data. All research was performed on patients in accordance with the Declaration of Helsinki of 1975, as revised in 2000, and the French regulations.

## 3. Results

Among the cohort of 259 consecutive patients admitted to the 6 ICUs between 10 March and 15 November 2020, 95 patients were excluded (92 did not receive mechanical ventilation and 3 with ESRD). Then, 164 consecutive patients were included in the final analysis (Figure 1).

Patients were mostly male (78.6%), overweight (65.9%), with a history of arterial hypertension (56.7%) and a mean age of 63 years. The mean initial APACHE-II score was 13. The median lengths of MV and ICU stay were 16.5 and 24 days, respectively. During their ICU stay, 42.7% displayed a septic shock and 16.5% required VV-ECMO. The mortality was 18.9% at D28 and 28.2% at D90 (Table 1). During the ICU stay, 97 patients (59.1%) developed an AKI, of which 39 severe stage 3 AKI (23.8%) and 21 (12.8%) required renal replacement therapy (RRT) for life-threatening indications (Figure 1).

In univariate analysis, stage 3 AKI was associated with ACEI exposure (*p* = 0.016), arterial hypertension (*p* = 0.029), APACHE II score (*p* = 0.004) and mortality at D28 (33.3% vs. 14.4%, *p* = 0.008), D60 (53.8 vs. 19.2%, *p* < 0.001) and D90 (56.4 vs. 19.2%, *p* < 0.001). These associations remained significant after the FDR-controlling procedure (adjusted *p*-values: *p* = 0.020, *p* = 0.029, *p* = 0.013, *p* < 0.001 and *p* < 0.001, respectively). Exposure to aminoglycosides (36.8% vs. 34.4%, *p* = 0.782), vancomycin (13.2% vs. 9.6%, *p* = 0.530) and iodinated contrast media (65.7% vs. 71.8%, *p* = 0.488) was not different between the two groups.

In multivariate analysis, the variables associated with the stage 3 AKI were: exposure to ACEI (OR: 4.238 (1.307–13.736), *p* = 0.016), APACHE II score (without age) (OR: 1.138 (1.044–1.241), *p* = 0.003) and iNO (OR: 5.694 (1.953–16.606), *p* = 0.001). The protective factors of stage 3 AKI were intubated prone positioning (OR: 0.234 (0.057–0.967), *p* = 0.045) and the use of dexamethasone (OR: 0.194 (0.053–0.713), *p* = 0.014) (Table 1).

PaO_2_/FIO_2_ ratio was not different in patients who received iNO and those who did not (*p* = 0.265). iNO was mainly prescribed for hypoxemia, as only 23.5% of the patients received iNO for right ventricular dysfunction. The mean duration and maximum dose of iNO were 6.4 days and 13.9 ppm, respectively. In univariate analysis, administration of iNO did not significantly affect mortality at D28, D60 and D90 (*p* = 0.398, *p* = 0.171, *p* = 0.106, respectively), but was also associated with RRT requirement (22.6% vs. 8.1%, *p* = 0.009). Dexamethasone was associated with reduced RRT requirement (4.2% vs. 16.2%, *p* = 0.038). Prone positioning was associated with lower mortality at D28 (16.1% vs. 33.3%, *p* = 0.036).

## 4. Discussion

In our cohort, we highlighted ACEI exposure, APACHE-II score (assessing initial gravity) and iNO were independently associated with severe AKI. Our study confirms the negative impact of AKI on mortality in critically ill patients with COVID-19 and the need to be careful about the risk factors of AKI and to implement strategies to limit nephrotoxicity in the ICU.

Meta-analyses suggested that iNO increases the risk of renal dysfunction by 40% in patients with ARDS, especially when receiving high cumulative doses [13], without reducing mortality [14]. The pathophysiology of iNO-induced AKI remains unclear, but the main hypothesis is the increased production of reactive oxygen species [15], exerting oxidative toxicity on the renal parenchymal cells. We report for the first time this independent association with both severe AKI and the need for RRT in severe COVID-19 patients. In our study, iNO did not improve mortality rates or the duration of MV and was associated with AKI and RRT. Thus, we suggest against its use in COVID-19-related ARDS when other therapeutic strategies are available.

Dexamethasone was associated with the prevention of AKI and RRT. This is consistent with a study by Orieux et al. highlighting that dexamethasone has an independent protective effect on AKI [16]. Indeed, dexamethasone could decrease glomerular and interstitial inflammation observed in kidneys, decreasing the risk of AKI during severe COVID-19, which may have contributed to a decrease in the AKI rate between the first and the second wave of COVID-19 pandemics [16]. Unexpectedly, prone positioning also seems to exert a protective effect on AKI. This could be explained by the improvement of lung compliance with prone positioning, allowing a less aggressive MV and a lower intra-thoracic pressure, therefore improving venous return and kidney congestion. In our institution, patients received both awake prone positioning before intubation (if tolerated) and after intubation in case of ARDS with PaO_2_/FiO_2_ ratio < 150 in patients already receiving neuromuscular blockers. The effects of prone positioning on kidneys are yet to be explored in the ongoing PRO-KIDNEY study (*NCT04286490*).

The main strength of our work is the exhaustivity of the multicentric data of consecutive patients from 6 ICUs within the same institution.

Our study has several limitations, mainly due to its retrospective design. We tried to take the selection bias into account, including both variables of global severity (age, APACHE-II score) and respiratory severity (PEEP and need for VV-ECMO), in our multivariate model. This cohort study was conducted during the first wave and the onset of the second wave of the COVID-19 outbreak in France [8]. Only a few patients received dexamethasone because its large use followed the results of RECOVERY published in July 2020 [7], and a few patients received Tocilizumab for the same reasons. Doses of iNO were heterogenous and ranged from 5 to 30 ppm, at the discretion of the clinician in charge. The number of patients receiving VV-ECMO was important in our cohort, as one of the ICUs is a regional referent centre for this technique. While a growing body of evidence suggests that AKI is frequent and associated with worse outcomes in patients receiving ECMO, we did not retrieve this association in our cohort.

## 5. Conclusions

In our cohort, dexamethasone was associated with the prevention of the risk of severe AKI and RRT, and iNO was associated with severe AKI and RRT in critically ill patients with COVID-19. This is further evidence of the anti-inflammatory effect of dexamethasone on the kidney during COVID-19. iNO should be used with caution in COVID-19-related ARDS, given its potential renal toxicity.

## Figures and Tables

**Figure 1 jcm-11-06130-f001:**
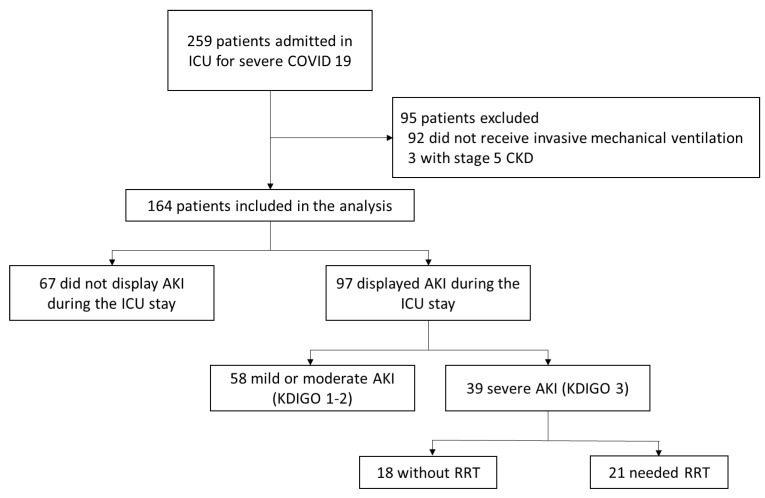
Flow chart of the study. AKI: Acute Kidney Injury, CKD: Chronic Kidney disease, ICU: Intensive Care Unit, RRT: Renal replacement therapy.

**Table 1 jcm-11-06130-t001:** Risk factors of stage III AKI in univariate and multivariate analysis. ACEI: Angiotensin-converting enzyme inhibitors, AKI: Acute Kidney Injury, ARB: Angiotensin receptor blocker, BMI: Body mass index, OR: Odds Ratio, ICU: Intensive Care Unit, MV: Mechanical ventilation, NSAID: Non-steroidal anti-inflammatory drugs, RRT: Renal replacement therapy, VV-ECMO: Veno-venous extra-corporeal membrane oxygenation. Bold: significant (*p* < 0.05).

	Total Population (n = 164)	KDIGO < 3 (n = 125)	KDIGO 3 AKI (n = 39)	*p* Univariate	OR (CI 95%)	*p* Multivariate
Age, years	62.9 ± 11.2	62.1 ± 11.0	65.5 ± 11.8	0.097	-	ns
Male, n (%)	126 (76.8)	95 (76.0)	31 (79.5)	0.652	-	-
BMI > 25 cm·Kg^2^	108 (65.9)	82 (65.6)	26 (66.7)	0.902	-	-
Arterial hypertension, n (%)	93 (56.7)	65 (52.0)	28 (71.8)	**0.029**	-	ns
Coronary arterial disease, n (%)	21 (12.8)	13 (10.4)	8 (20.5)	0.099	-	ns
Chronic heart failure, n (%)	4 (2.4)	3 (2.4)	1 (2.6)	1.000	-	-
Severe cardiovascular disease, n (%)	34 (20.7)	23 (18.4)	11 (28.2)	0.187	-	ns
Diabetes mellitus, n (%)	62 (37.8)	43 (34.4)	19 (48.7)	0.107	-	ns
Chronic respiratory disease, n (%)	28 (17.1)	18 (14.4)	10 (25.6)	0.103	-	ns
Chronic kidney disease, n (%)	11 (6.7)	4 (3.2)	7 (17.9)	**0.004**	-	ns
- Stage 1	1 (0.6)	1 (0.8)	0 (0.0)	>0.999		
- Stage 2	3 (1.8)	2 (1.6)	1 (2.6)	0.560		
- Stage 3	5 (3.0)	1 (0.8)	4 (10.3)	**0.012**		
- Stage 4	2 (1.2)	0 (0.0)	2 (5.1)	0.055		
NSAID, n (%)	4 (2.4)	3 (2.4)	1 (2.6)	1.000	-	-
ACEI, n (%)	26 (15.9)	15 (12.0)	11 (28.2)	**0.016**	**4.238 (1.307–13.736)**	**0.016**
ARB, n (%)	28 (17.1)	19 (15.2)	9 (23.1)	0.254	-	-
APACHE II, mean ± SD	13.0 ± 6.1	11.9 ± 5.2	16.3 ± 7.8	**0.004**		
APACHE II without age points, mean ± SD	9.2 ± 5.8	8.3 ± 5.1	12.1 ± 7.0	**0.006**	**1.138 (1.044–1.241)**	**0.003**
Septic shock, n (%)	70 (42.7)	52 (41.6)	18 (46.2)	0.616	-	-
PEEP, cmH_2_O	12.7 ± 3.1	12.7 ± 3.1	12.6 ± 3.3	0.896	-	ns
Plateau pressure, cmH_2_O	25.1 ± 4.8	25.0 ± 5.1	25.3 ± 3.9	0.729	-	-
Compliance _RS_, mL/cm H_2_O	36.9 ± 12.2	37.0 ± 12.7	36.7 ± 10.3	0.918	-	-
Driving pressure, cmH_2_O	12.1 ± 4.2	12.0 4.3	12.5 ± 3.9	0.613	-	-
PaO_2_/FIO_2_ ratio	141.8 ± 59.1	142.3 ± 61.2	140.3 ± 53.0	0.859	-	-
Prone positioning (intubated), n (%)	137 (83.6)	108 (86.4)	29 (74.4)	0.077	**0.234 (0.057–0.967)**	**0.045**
Awake prone positioning, n (%)	22 (13.4)	19 (15.2)	3 (7.7)	0.230	**-**	**-**
Inhaled Nitric Oxide, n (%)	53 (32.3)	36 (28.8)	17 (43.6)	0.085	**5.694 (1.953–16.606)**	**0.001**
Duration of iNO, days	6.4 ± 4.6	6.4 ± 4.5	6.3 ± 5.0	0.943	**-**	**-**
Maximum dose, ppm	13.9 ± 5.5	12.1 ± 3.4	14.8 ± 6.2	0.080	**-**	**-**
iNO for right ventricular dysfunction	12 (23.5)	7 (19.4)	5 (33.3)	0.287		
VV-ECMO, n (%)	27 (16.5)	18 (14.4)	9 (23.1)	0.202	-	ns
Neuromuscular blockage, n (%)	156 (95.1)	119 (95.2)	37 (94.9)	1.000	-	-
Dexamethasone, n (%)	47 (28.7)	40 (32.0)	7 (17.9)	0.090	**0.194 (0.053–0.713)**	**0.014**
Hydroxychloroquine, n (%)	106 (64.6)	82 (65.6)	22 (61.5)	0.643	-	-
Azithromycin, n (%)	33 (67.3)	26 (65.0)	7 (77.8)	0.460	-	-
Lopinavir/ritonavir, n (%)	18 (11.0)	13 (10.4)	5 (12.8)	0.673	-	-
Tocilizumab, n (%)	3 (1.8)	2 (1.6)	1 (2.6)	0.695	-	-
Ruloxitinib, n (%)	19 (11.6)	13 (10.4)	6 (15.4)	0.396	-	-
Anakinra, n (%)	18 (11.0)	13 (10.4)	5 (12.8)	0.673	-	-
Aminoglycosides, n (%)	57 (35.0)	43 (34.4)	14 (36.8)	0.782	-	-
Vancomycin, n (%)	17 (10.4)	12 (9.6)	5 (13.2)	0.530	-	-
Iodinated contrast media, n (%)	112 (70.4)	89 (71.8)	23 (65.7)	0.488	-	-
Higher serum urea level mmol/L	19.8 ± 12.4	15.0 ± 7.2	36.6 ± 12.1	**<0.001**	-	-
Higher serum creatinine level, µmol/L	163.8 ± 139.3	106.3 ± 48.2	363.4 ± 166.5	**<0.001**	-	-
**Outcome**						
D28 mortality, n (%)	31 (18.9)	18 (14.4)	13 (33.3)	**0.008**	-	-
D60 mortality, n (%)	45 (27.6)	24 (19.2)	21 (53.8)	**<0.001**	-	-
D90 mortality, n (%)	46 (28.2)	24 (19.2)	22 (56.4)	**<0.001**	-	-
RRT, n (%)	21 (12.8)	0 (0.0)	21 (53.8)	**<0.001**	-	-
Length of stay in ICU, days	24 (15–43)	23 (15–41)	29 (17–46)	0.479	-	-
Length of stay in hospital, days	34 (21–49)	32.5 (21–47)	37 (17–49)	0.827	-	-
Length of MV, days	16.5 (8–32)	16 (8–31)	18 (9–37)	0.578	-	-

## Data Availability

The datasets used and analysed during the current study are available from the corresponding author on reasonable request.

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
