# Peer review of "Impact of Dexamethasone and Inhaled Nitric Oxide on Severe Acute Kidney Injury in Critically Ill Patients with COVID-19"

_jcm, 2022, doi:10.3390/jcm11206130_

Round 1

Reviewer 1 Report

The study from Mickaël Bobot and colleagues aimed to evaluate the efficacy of corticotherapy and iNO in critical COVID-19 patients with AKI. Since kidney injury is one of the main complications observed in the ICU for COVID-19 patients, and directly impacts on mortality, this study adds significant knowledge. Here are my considerations:

- The sentence from the lines 57-58 should be referenced.

- The abbreviation “VV-ECMO” should be defined in line 88, when it first appears.

- As a suggestion… Since 2018, KDIGO recommends to use the term “kidney failure” instead of “ESRD” since it may be sensitive to patients. I also suggest to include the cut-off for GRF that was considered as kidney failure and as an exclusion criterion.

- In the same topic: CKD patients included in this study were G2-G3b or G2-G4? I suggest to include this information on Table 1.

- I believe that the authors did not find any relevant risk/protective factors for AKI stages 1 and 2. I suggest to include a sentence to clarify this.

Author Response

Reviewer #1:

The study from Mickaël Bobot and colleagues aimed to evaluate the efficacy of corticotherapy and iNO in critical COVID-19 patients with AKI. Since kidney injury is one of the main complications observed in the ICU for COVID-19 patients, and directly impacts on mortality, this study adds significant knowledge. Here are my considerations:

  1. The sentence from the lines 57-58 should be referenced.

è As suggested, we added the reference of Ferlicot et al. Nephrol Dial Transplant 2021 about the spectrum of kidney biopsies and slightly modified the sentence to: “Various COVID-19-related kidney impairments were described, but the most frequent feature is acute tubular necrosis with interstitial inflammatory involvement, followed by glomerular pattern like focal segmental glomerulosclerosis“, to be more accurate.

  1. The abbreviation “VV-ECMO” should be defined in line 88, when it first appears.

è We agree and have modified as suggested.

  1. As a suggestion… Since 2018, KDIGO recommends to use the term “kidney failure” instead of “ESRD” since it may be sensitive to patients. I also suggest to include the cut-off for GRF that was considered as kidney failure and as an exclusion criterion.

è Thank you for this remark. We modified changed the term of ESRD to “stage 5 CKD (kidney failure)” in the manuscript and Figure 1 in order to avoid any confusion. The cut-off of GFR defining stage 5 CKD is below, 15 ml/min/1.73m², which we added in the manuscript as you suggested.

  1. In the same topic: CKD patients included in this study were G2-G3b or G2-G4? I suggest to include this information on Table 1.

è Thank you for this suggestion. We included CKD patients from stage 1 to stage 4. This had been detailed in table 1, as suggested.

  1. I believe that the authors did not find any relevant risk/protective factors for AKI stages 1 and 2. I suggest to include a sentence to clarify this.

  • Indeed we did not find relevant factors associated with mild and moderate AKI. We chose to focus on stage 3 AKI which have the most important negative impact on mortality, and as less severe AKI can be associated with many confounders, especially in the beginning of the hospitalization (i.e. diarrhea etc.). We have specified it in the methods section as suggested by the reviewer.

Reviewer 2 Report

The clinical question is important; however, the causal inference between dexamethasone and inhaled nitric oxide on severe 2 acute kidney injury cannot be fully established due to the retrospective study design. More sophisticated methodology of causal inference can be applied. 

1. "their impact on kidney function was not assessed in the COVID-19 patients"---AKI can be a bystander of hypoxia, more severe ARDS will be associated with more severe AKI. So are there any direct causal link between inhaled nitric oxide  and AKI? it might be via hypoxia. did you consider mediation analysis to reveal the causality. 

2. There can be many types of steroids in clinical practice. Also there are many such reports in the literature, you need to acknowledge these background.

3. why not consider PSM to balance baseline characteristics of the treated and control groups? 

4. There are many statistical tests, which overinflated type I error. 

5. iNO showed a very high OR, which can be due to selection bias. 

6. prone positioning is found to be helpful, but what is the timing of e prone positioning? some authors suggest awake prone positioning before intubation. 

7. "PaO2/FiO2 ratio was not different in patients who received iNO"--Are there any other factors that can determine who will receive iNO? 

Author Response

Reviewer #2: 

The clinical question is important; however, the causal inference between dexamethasone and inhaled nitric oxide on severe 2 acute kidney injury cannot be fully established due to the retrospective study design. More sophisticated methodology of causal inference can be applied. 

  1. "their impact on kidney function was not assessed in the COVID-19 patients"---AKI can be a bystander of hypoxia, more severe ARDS will be associated with more severe AKI. So are there any direct causal link between inhaled nitric oxide and AKI? it might be via hypoxia. did you consider mediation analysis to reveal the causality. 

Thank you for this insightful remark. Mediation analysis could be really interesting. Unfortunately, we do not master mediation analysis, and won’t be able to address specialized statistical analyses in the short time allowed by the journal for revision. We addressed these limitations in the Discussion section. However, we included VV-ECMO and PEEP, as variables of respiratory severity in our multivariate model to limit this bias.

  1. There can be many types of steroids in clinical practice. Also there are many such reports in the literature, you need to acknowledge these background.

The reviewer is right. We developed the different steroids use in critical care and added the following sentences in the Introduction session: “Corticosteroids are widely used in critical care for their anti-inflammatory and immunosuppressive proprieties, and have proven efficiency in various conditions like anaphylaxis, septic shock, exacerbation of asthma or post-aggressive pulmonary fibrosis (Young et al., 2018). There are many types of corticosteroids used in clinical practice with varying glucocorticoid and mineralocorticoid effects. Since July 2020 and the results of RECOVERY study, dexamethasone was shown to reduce the mortality in COVID-19 patients”.

  1. why not consider PSM to balance baseline characteristics of the treated and control groups? 

Thank you for this important suggestion. Propensity score matching may allow to overcome the selections bias. However, its main limitation is the loss of size and therefore of statistical power. In our study of limited size, after PSM on age, APACHE-II score and ECMO, the groups become comparable but of small size (because there are only 39 patients in the KDIGO 3 group) and not very representative of our intensive care population. We obtain outlier results, notably for iNO (see the Tables below).

KDIGO < 3 (n=34)

KDIGO 3 AKI

(n=34)

p

Age, years*

62.4 ± 1.7

65.5 ± 2.1

0.333

Male, n (%)

25 (73.5)

27 (79.4)

0.567

Arterial Hypertension, n (%)

17 (47.1)

23 (70.6)

0.085

Coronary artery disease, n (%)

4 (11.8)

7 (20.6)

0.512

Severe cardiovascular disease, n (%=

9 (26.5)

9 (26.5)

0,999

Diabetes mellitus, n (%)

11 (32.4)

16 (47.1)

0.215

Chronic respiratory disease, n (%)

6 (17.6)

9 (26.5)

0.560

REIN

1 (2.9)

6 (17.6)

0.105

APACHE II, mean ± ESM

13.3 ± 1.2

16.2 ± 1.3

0.210

Septic shock, n (%)

18 (52.9)

15 (44.1)

0.467

PEEP, cmH2O, mean ± ESM

11.5 ± 0.4

12.6 ± 0.6

0.115

PaO2/FIO2 ratio

129.4 ± 11.2

141.3 ± 9.1

0.413

Prone Positioning (intubated), n (%)

32 (94.1)

25 (73.5)

0.021

Inhaled Nitric Oxide, n (%)

27 (79.4)

15 (44.1)

0.003

VV-ECMO, n (%)

10 (29.4)

7 (20.6)

0.401

Dexamethasone, n (%)

12 (35.3)

6 (17.6)

0.099

ACEI, n (%)

6 (17.6)

10 (29.4)

0.202

Lopinavir/ritonavir, n (%)

5 (14.7)

4 (11.8)

0.720

D28 Mortality, n (%)

8 (23.5)

12 (35.3)

0.287

*mean, ± ESM,

Supplementary Table 1 : Risk factors of stage III AKI after matching on Age, APACHE II, and VV-ECMO . ACEI: Angiotensin Converting Enzyme Inhibitors, RRT: Renal replacement therapy, VV-ECMO: Veno-venous extra-corporeal membrane oxygenation

Supplementary Table 2. Propensity score matching - caliper 0.2

KDIGO <3

(n=34)

KDIGO≥3

(n=34)

p

SMD

Age, years

62.4 ± 1.7

65.5 ± 2.1

0.333

0.268

APACHE II

13.3 ± 1.2

16.2 ± 1.3

0.210

0.401

VV-ECMO

10 (29.4)

7 (20.6)

0.401

0.204

SMD: Standardized mean difference

  1. There are many statistical tests, which overinflated type I error. 

Thank you for this remark. We agree with you on this pointWe performed a Benjamini-Hochberg False Discovery Rate controlling procedure for multiple testing to limit the risk of overinflation of type I error. The comparisons remained significant. This was added to the Methods and Results sections, with adjusted p-values. The results of the performed tests are coherent with our clinical hypotheses about notably iNO and dexamethasone and AKI in COVID patients.

  1. iNO showed a very high OR, which can be due to selection bias. 

The reviewer is right, we initially thought that this may be due to global severity of patients receiving iNO, especially during the two first waves of COVID-19 pandemics. We tried to take the selection bias into account by including both variables of global severity (age, APACHE-II score) and respiratory severity (PEEP and need for VV-ECMO) in our multivariate model. We specified it in the discussion section.

  1. prone positioning is found to be helpful, but what is the timing of e prone positioning? some authors suggest awake prone positioning before intubation. 

We reported data from sedated prone positioning. In our institution, patients received both awake prone positioning before intubation (if tolerated), and after intubation in case of ARDS with PaO2/FiO2 ratio < 150 in patient already receiving neuromuscular blockers. We added it in the Discussion section. We added information about awake prone positioning, which seem to have no impact on severe AKI. Unfortunately, we do not have information on the precise timing of the prone positioning in relation to the AKI, however the information about prone positioning was recorded before the AKI occurred or concomitantly (in the first 72 hours of the AKI for patients with an initial AKI).

  1. "PaO2/FiO2 ratio was not different in patients who received iNO"--Are there any other factors that can determine who will receive iNO? 

Thank you for raising this important point. The other factor who can determine who will receive iNO is the presence of a right ventricular dysfunction. In our cohort, iNO was prescribed for right ventricular dysfunction in 23.5% of the patients. The proportion of iNO prescription in this indication was not different between the Stage 3 AKI and control groups (33.3% vs. 19.4% p=0.287). We added these data in the Results section and the Table 1.

Reviewer 3 Report

The impact of nephrotoxic antibiotics as well as the role of contrast imaging on the development of AKI should be shown

Author Response

Reviewer #3:

The impact of nephrotoxic antibiotics as well as the role of contrast imaging on the development of AKI should be shown.

è Thank you for your insightful suggestion. We found no impact of the nephrotoxic antibiotics nor iodinated contrast media on the development of AKI (we think due to strict monitoring of the residual blood concentrations). We added data about iodinated contrast media in Table 1 and the following sentence in the Results section of the manuscript: “Exposure to aminoglycosids (36.8% vs. 34.4%, p=0.782), vancomycin (13.2% vs. 9.6%, p=0.530) and iodinated contrast media (65.7% vs. 71.8%, p=0.488) was not different between the 2 groups.”

Round 2

Reviewer 2 Report

My previous comments are well addressed

Author Response

We thank the reviewer for her/his insightful suggestions that importantly helped to improve our manuscript.